# SDS Depletion from Intact Membrane Proteins by KCl Precipitation Ahead of Mass Spectrometry Analysis

**DOI:** 10.3390/proteomes13030030

**Published:** 2025-07-02

**Authors:** Tania Iranpour, Mapenzi Mirimba, Chloe Shenouda, Adam Lynch, Alan A. Doucette

**Affiliations:** Department of Chemistry, Dalhousie University, 6243 Alumni Crescent, Halifax, NS B3H 4R2, Canada; tn315509@dal.ca (T.I.); mp633299@dal.ca (M.M.); ch976052@dal.ca (C.S.); ad647601@dal.ca (A.L.)

**Keywords:** sodium dodecyl sulfate, detergents, protein purification, intact proteins, KCl precipitation, membrane proteomics, hydrophobic proteins, sample preparation, mass spectrometry

## Abstract

Background: Membrane proteins are preferentially solubilized with sodium dodecyl sulfate (SDS), which necessitates a purification protocol to deplete the surfactant prior to mass spectrometry analysis. However, maintaining solubility of intact membrane proteins is challenged in an SDS-free environment. SDS precipitation with potassium salts (KCl) offers a potentially viable workflow to deplete SDS and permit proteoform analysis. The purpose of this study is to devise a robust detergent-based protocol applicable for processing and analysis of intact membrane-associated proteoforms. Methods: The precipitation conditions impacting SDS removal from spinach chloroplasts and liver membrane proteome preparations were evaluated, capitalizing on optimization of pH (highly basic), addition of MS-compatible solubilizing additives (urea) and adjustment of the KCl to SDS ratio to maximize recovery and purity. Results: Characterization of the SDS-solubilized, KCl-precipitated spinach membrane preparation revealed multiple charge envelope MS spectra displaying high signal to noise, free of SDS adducts. Precipitation at pH 12 or with urea improved protein recovery and purity. Bottom-up analysis identified 1826 distinct liver protein groups from four independent SDS precipitation conditions. While precipitation at pH 8 without urea revealed a greater number of protein identifications by mass spectrometry, precipitation under highly basic conditions (pH 12) with urea provided higher membrane protein recovery and achieved the greatest number (732 of 1056) and largest percentage (69.3%) of membrane proteins identified in the SDS removal workflow. Conclusion: This workflow provides new opportunities for MS-based proteoform analysis by capitalizing on the benefits of SDS for protein extraction while maintaining high solubility and purity of intact proteins though KCl precipitation of the surfactant.

## 1. Introduction

Sodium dodecyl sulfate (SDS) is a favored surfactant in proteomics, primarily to facilitate cell lysis, extraction, and solubilization of hydrophobic membrane components which are difficult to recover in a purely aqueous solution [1]. Unfortunately, even modest concentrations of SDS will significantly compromise downstream protein processing and analysis, including inhibition of trypsin digestion [1], deterioration of reversed phase separation [2], and suppression of electrospray ionization for mass spectrometry detection [3]. Multiple SDS depletion strategies exist to lower the surfactant concentration to acceptable levels prior to downstream processing, emphasizing a critical threshold of 0.01% SDS (~35 mM) that is needed to minimize undesirable impacts for digestion, separation, and MS detection [3]. Naturally, the removal of SDS from proteins should also coincide with maintaining high protein recovery.

A summary listing of strategies for SDS removal is provided in Table 1, including some commercial options. Certain strategies are not applicable to intact proteome analysis, as the final product of the workflow is digested peptide fragments. Others are best suited for peptide level processing, as intact proteoform recovery and/or purity may be compromised. Among the depletion strategies are methods that capture the protein, releasing the detergent-containing solution. Cartridge formats such as FASP [4], Strap [5] and sp3 [6], as well as solvent-based protein precipitation [7] are examples that employ this strategy. Precipitation strategies in particular are not only simplistic and affordable to use, but can provide exceptional throughput [8], while achieving the desired purity and high protein yield [7]. Perhaps most important, they are suitable to process intact proteins, and are a favored component of multiple detergent-based workflows permitting proteoform processing by top-down MS [9]. One limitation of protein precipitation, however, is that the resulting pellet must be resolubilized, which introduces additional processing steps, lowering throughput, and potentially contributes biased loss of critical proteoform components.

Capitalizing on precipitation-based workflows, protein purification by direct precipitation of the SDS would, theoretically, retain analytes in solution, thus alleviating subsequent protein resolubilization steps [17]. While the sodium salt, SDS, is highly soluble at up to 20% by weight in water, potassium dodecyl sulfate (KDS) has limited solubility of only 415 mg/L (~1.4 mM) at 25 °C. To that end, multiple reports have described the addition of potassium salts to precipitate SDS from proteins. As early as 1973, van Heyningen demonstrated that excess K^+^ ions would deplete free SDS, allowing proteins to remain in solution [18]. However, the strong association between SDS and protein must be considered as they can impact the efficiency of the process [19]. SDS binds to proteins electrostatically, whereby the negative head group of the surfactant attracts to positively charged basic residues of the protein at a low pH; SDS-bound protein can even adopt a structure of amyloid fibrils [20]. At a neutral or high pH, hydrophobic attractions between SDS and aromatic or non-polar side chains induce α-helical structures, occurring at up to a 20:1 mol ratio of SDS to protein [20]. Interactions between SDS and protein challenge the opportunity for potassium-induced SDS precipitation in an MS-based proteomics workflow, as there exists a potential for sample loss due to co-precipitation of proteins, and/or high residual surfactant remaining in the sample.

In 2012, Zhou et al. employed KCl precipitation of SDS for MS-based proteome analysis [21]. They reported over 99.99% SDS depletion, with greater than 96% peptide recovery. A limitation of this study, however, was that the SDS depletion step was conducted at the peptide level, following tryptic digestion. Top-down MS would not be possible. Moreover, for the enzyme to remain active, the initial SDS concentration must be minimal (below 0.05%). Zhou’s study did not report the pH of their detergent precipitation protocol, though trypsin-digested samples are typically acidified to quench the reaction.

Several studies have alluded to sample pH as a controlling variable affecting SDS–protein interactions during K^+^-induced SDS precipitation. Sandri et al. described a method to precipitate only the ‘free’ SDS from proteins using room temperature incubation with 180 mM KCl at pH 12; SDS-bound proteins remained in solution [22]. Carraro et al. added 10% TCA to a pH < 3, along with 180 mM KCl and incubation on ice, resulting in precipitation of the SDS-bound proteins [19]. Li et al. boiled proteins in 1% SDS along with 0.5 M KCl at pH 8, followed by ice treatment to induce protein precipitation, and claiming greater than 95% protein recovery in the pellet [23]. Higher temperatures decrease the aggregation number and disrupt larger detergent micelles [24,25]. Takedo et al. incubated protein and SDS with potassium ions at neutral pH, demonstrating a strong dependence of residual SDS on the potassium ion concentration [26]. They observed a lower plateau following the addition of 300 mM K^+^ to 0.5% initial SDS. Nearly all SDS was removed, including most of the protein-bound surfactant. Takedo’s studies were conducted with BSA and chymotrypsin, although MS examination of purified proteins was not performed. In 2018, Žilionis et al. examined multiple SDS depletion strategies prior to MS analysis of the purified proteins [27]. They reported that KCl precipitation at pH 7.4 resulted in the lowest recovery of intact BSA (62.25%), relative to all other SDS depletion methods, including FASP, and solvent-based protein precipitation.

Despite the prominence of potassium-induced SDS precipitation in the literature, to date, no study has employed this strategy for purification of intact proteoform mixtures ahead of MS analysis. While BSA or other water-soluble proteins serve as potentially useful models to examine SDS depletion efficiency, such systems do not require SDS for proteoform extraction. By contrast, a membrane proteome preparation necessitates solubilizing additives. To that end, we herein consider the addition of alternative solubilizing additives, namely urea, as well as high pH conditions, to maintain the solubility of intact proteins during and following SDS depletion from a membrane proteome preparation. This study therefore addresses a critical knowledge gap associated with potassium-induced SDS depletion. Our optimization of recovery and purity for intact membrane proteins permits in-depth proteomics analysis by LC-MS, providing researchers with a robust alternative workflow to process SDS-containing proteomic mixtures, and permit proteoform characterization through bottom-up or top-down mass spectrometry.

## 2. Materials and Methods

### 2.1. Chemicals

Bovine serum albumin (BSA, ≥98%), TPCK-treated trypsin (cat. T1426), bovine cytochrome c, ubiquitin and ribonuclease B, equine myoglobin, porcine pepsin, urea, sodium carbonate (≥99.5%), sodium hydroxide (≥98%), and methylene blue (≥97%) were obtained from MilliporeSigma (Oakville, ON, Canada). KCl (≥99%) was from ACP Chemicals (Montreal, QC, Canada). The Pierce^TM^ BCA Protein Assay Kit, Tris (≥99.9%), and iodoacetamide (≥98%) were sourced from Thermo Fisher Scientific (Waltham, MA, USA). Sodium dodecyl sulfate (SDS) and dithiothreitol (DTT) were from Bio-Rad (Hercules, CA, USA). Chloroform (≥99.9%), methanol (HPLC Grade, ≥99.9%), and trifluoroacetic acid (TFA, HPLC Grade, 98%) were supplied by Fisher Scientific (Hampton, NH, USA). Distilled water was purified to 18.2 MΩcm using a Milli-Q purifier. Yeast (*Saccharomyces cerevisiae*), spinach, and bovine liver were obtained from a local grocery store. All other materials were used without further purification.

### 2.2. Total Proteome Extraction

Partially thawed liver was shaved into ~1 mm slices, frozen in liquid nitrogen, then ground to a fine powder in a coffee grinder. The total liver protein extraction was obtained by adding 50 mM aqueous Tris buffer (pH ~8) at a mass ratio of 5:1 (buffer to liver). The mixture was sonicated on ice for 1 h, followed by centrifugation at 21,000× *g* for 5 min. The supernatant was retained, filtered, and stored at −20 °C. Similarly, cultured *Saccharomyces cerevisiae* cells grown overnight in yeast–peptone–dextrose (YPD) broth at 30 °C until OD600 ≤ 1, were harvested and washed with cold 50 mM Tris buffer (pH 8), resuspended in said buffer at a 2:1 mass ratio (buffer to yeast), then ground to a fine powder in a mortar and pestle. The resulting powder was further diluted 5:1 with ice-cold Tris buffer, sonicated on ice for 1 h, and centrifuged at 21,000× *g* for 5 min to obtain the final total protein extract.

### 2.3. Membrane Proteome Preparations

The membrane proteome preparation involves pretreatment with 0.1 M sodium carbonate buffer, pH 11, by incubating on ice for 30 min at a 1:1 ratio [28]. The mixture was briefly sonicated, then spun at 100,000 rpm, in a TLA−100.3 fixed angle rotor with a Beckman Optimal TL Ultracentrifuge (Indianapolis, IN, USA) for 1 h at 4 °C. The resulting pellet was resuspended in a minimal volume of ice-cold Tris buffer, centrifuged once more using the aforementioned conditions, before resuspending the final pellet in 2% SDS. From a BCA protein assay of the clarified supernatant, proteins in the membrane fraction were diluted in 2% SDS to a working concentration of 2 g/L and stored at −20 °C.

Spinach chloroplast membranes were prepared by homogenizing 50 g spinach leaves (*Spinacia oleracea*) in a blender (10 s) with 100 mL of 10 mM Tris (pH 7.5), 0.33 M sucrose, and 2 mM EDTA, which was then filtered through cheese cloth. The supernatant was centrifuged (1000× *g*, 10 min, 4 °C) and the resulting pellet was resuspended in 10 mM Tris (pH 7.5), 2 mM EDTA to disrupt the cells, then centrifuged at 4300× *g* (10 min, 4 °C). The resulting membrane pellet was solubilized at 20% *w*/*v* in 2% SDS, then subject to methanol/chloroform/water precipitation (4:1:3 parts by volume) to isolate membrane proteins. The purified protein was resolubilize in 0.5% SDS (with or without 8 M urea, pH 8 or 12) and subject to KDS precipitation.

### 2.4. KDS Precipitation

The efficiency of KDS precipitation was evaluated with respect to solution pH, temperature, KCl to SDS ratio, inclusion of urea, and for different protein sources (standard proteins, liver sample, spinach membranes, liver membrane preparation, and yeast membrane preparation). All reported concentrations of SDS, protein, urea, buffer, and KCl are for the solutions following addition of KCl, which occurred in a 1:1 ratio with the sample. Solid urea was added by mass to the buffered SDS–protein stocks when required, to arrive at the final desired concentration. The pH of the precipitating solution was controlled using sodium citrate (50 mM final), sodium carbonate (50 mM final), or sodium hydroxide buffers to achieve final solution pH values ranging from 3 to 13. Sample temperatures were maintained in a warm water bath, or a chilled centrifuge (without spinning), preset to the specified temperature. A Design of Experiments (DoE) approach was completed to examine the combined impact of KCl to SDS ratio (10:1 or 50:1), pH (4.5, 8, 13), and urea (0 or 7.5 M).

Following the addition of KCl, samples were immediately vortexed, then incubated at a specified temperature without perturbation for 30 min. Samples were next centrifuged for 20 min at 13,000× *g* to pellet the resulting precipitate. To avoid accidental pipetting of the surfactant pellet, only the upper 50% of the supernatant was retained, being subject to residual SDS quantitation and protein recovery. Samples were also subject to trypsin digestion and subsequent LC-MS/MS.

### 2.5. SDS Quantification

The methylene blue active substances (MBASs) assay was employed following the guidelines reported by Arand et al. [29]. Samples were first diluted 10-fold in water, to which the methylene blue reagent (250 mg methylene blue, 50 g sodium sulfate, 10 mL concentrated sulfuric acid per liter aqueous) was combined in a 1:1 ratio, followed by a 2:1 addition of chloroform. After brief vortexing and centrifugation, the upper layer was removed by pipetting and the chloroform layer was quantified by measuring absorbance at 652 nm. Calibration curves of SDS over the concentration range 5 to 25 ppm, matrix matched to each of the experimental conditions with respect to buffer, urea, and KCl concentration, were constructed and used to translate absorbance data to residual SDS.

### 2.6. Peptide/Protein Recovery Determination

Protein quantification was performed using the bicinchoninic acid (BCA) assay as well as HPLC-UV. In the BCA assay, 50 µL of each sample was incubated with 300 µL of BCA working reagent at 56 °C for 30 min. Absorbance was measured at 562 nm and compared to a standard curve of BSA ranging from 0.05 to 0.1 g/L, matrix-matched to the test samples being measured. LC-UV quantitation and sample cleanup involved generating a calibration curve of intact or trypsin-digested BSA, injecting from 1 to 15 µg into a 1 × 50 mm self-packed reversed phase column, made with Poros 20 R2 beads (Thermo Fisher Scientific). Peptides or protein eluted as a single peak and were collected as a single fraction by employing a stepwise gradient from 5 to 50% acetonitrile in 0.1% TFA water and monitoring the UV absorbance at 214 nm [30].

### 2.7. SDS-PAGE

Prior to loading into the gel, all samples were subject to reversed phase cleanup, to reduce the potential impacts of high salt or urea on SDS PAGE. Cleaned samples were dried in a SpeedVac (Thermo Fisher Scientific) before being resolubilized in 25 µL Laemmli buffer (96 °C, 5 min). Samples were loaded onto a 12% SDS-PAGE gel, resolved at 130 V, and stained with Coomassie Blue following standard protocols.

### 2.8. Protein Digestion

The supernatant isolated following KDS precipitation of 1 g/L liver membrane proteins was diluted five-fold, then subject to trypsin digestion, employing 100 µL of SDS-depleted protein. The pH was adjusted to 8.3 (50 mM Tris), after which proteins were reduced using 5 mM DTT at 56 °C for 30 min. Cysteine residues were alkylated with 11 mM iodoacetamide at room temperature in the dark for 30 min. Trypsin was added at a 1:20 mass ratio (trypsin to protein), and samples were incubated overnight at 37 °C. The digestion was quenched by acidifying in 1% TFA. The final protein concentration was 0.2 µg/µL (assuming 100% recovery throughout precipitation and digestion). Digested samples were next desalted by LC/UV cleanup, dried by SpeedVac, and stored at −20 °C prior to LC-MS/MS analysis.

### 2.9. LC-MS Analysis

Digested protein samples were injected into a Dionex Ultimate 3000 LC nanosystem (Bannockburn, IL, USA) system, using a self-packed 75 µm × 25 cm C18 column with integrated nanospray tip packed with 1.9 *μ*m Reprosil-Pur Basic beads from Dr. Maisch, Ammerbuch, Germany). A 1 hr linear gradient progressed from 0.1% formic acid in water to 40% acetonitrile. The Orbitrap Lumos mass spectrometer (Thermo Fisher Scientific) operated in the data-dependent mode over the mass range 375−1400 *m/z*, with MS^1^ at a resolution of 60,000 FWHM, and MS^2^ scanned at 30,000 resolution using high-energy CID set at 30%. A 3 s cycle time was employed, with 60 s dynamic exclusion (±20 ppm). Other settings included Nanospray source voltage 2.4 kV; automatic gain control of 4 × 10^5^ in MS^1^, 1 × 10^5^ in MS^2^; ion transfer tube temperature of 275 °C; RF lens at 45%; default charge state +2; MS^2^ isolation window of 1.6 *m/z*.

Intact proteins were analyzed by LC-MS by injecting the KCl-precipitated samples onto a self-packed 0.5 mm × 25 cm C18 column packed with 3 μm BioBasic beads (Phenomenex, Torrance, CA, USA). There was a 1 hr linear gradient from 5% to 70% acetonitrile in water with 0.1% formic acid, at a column flow rate of 12 µL/min. For standard proteins, the column was connected via an electrospray source (3.5 kV) to a Bruker Compact Q-TOF (Billerica, MA, USA). Spinach membranes were analyzed on an Orbitrap Q Exactive (Thermo Fisher Scientific) operating over the mass range 500−2000 *m/z* at a resolution of 60,000 FWHM, electrospray voltage 3.5 kV, automatic gain control 1 × 10^6^, ion transfer tube temperature 300 °C and RF lens at 60%.

### 2.10. Data Analysis

MS spectral data were searched using MaxQuant (v. 2.4.9.0) against the *Bos taurus* database, downloaded from UniProt on 09/03/2024, and containing 37,501 entries. MS1 tolerance was set to 20 ppm and 5 ppm for MS2. Methionine oxidation was set as a variable modification, with carbamidomethylation of cysteine as a fixed modification. A peptide false discovery rate of 0.01 was selected, with up to two missed cleavages. A protein charge state calculator was generated in Excel, employing pKa values of isolated amino acid residues (data from Harris & Lucy, Quantitative Chemical Analysis, 10th Edition, 2020, [31]). The average charge of the liver proteome employed the relative abundance of amino acids from Bos taurus, with an average protein length of 600 residues. Intact protein MS spectra were deconvoluted using Excel. Design of Experiments regression analysis was performed using the built-in tools in Excel.

## 3. Results

### 3.1. Peptide Level vs. Protein Level SDS Purification

KCl precipitation of SDS has previously been employed to purify trypsin-digested peptides, permitting bottom-up MS-based proteomics processing [14]. While Zhou’s study did not explore the influence of pH on peptide recovery during precipitation, it is anticipated to impact analyte yield. Test samples were subject to overnight trypsin digestion, spiked with 0.5% SDS (post-digest) and thereafter subjected to KCl precipitation at varying pH values. When employing highly basic conditions (pH 13), any potential ionic interactions between SDS and peptide are eliminated as basic residues are deprotonated. Under these conditions, Figure 1A confirms 96 ± 3% and 98 ± 1% average peptide recovery for digested BSA and bovine liver (total proteome extract), respectively. Retaining the digestion buffer pH of 8 during precipitation caused only a modest drop in recovery of liver peptides (89 ± 3%), and no significant change for BSA peptide recovery (95 ± 2%). However, lowering the pH to 3 resulted in a recovery decrease to 67 ± 1% for both the digested BSA and liver peptides. Typically, trypsin is quenched in acid, so it would be important to omit this step when employing KCl to precipitate SDS. Nonetheless, SDS removal at the peptide level implies the presence of surfactant during digestion; a concentration above 0.1% SDS would deactivate trypsin. We previously reported that an SDS concentration as low as 0.01% has a detrimental impact on cumulative enzyme activity, leading to a reduction in total peptide identifications and therefore sub-optimal MS-based proteome analysis [28].

Alternatively, SDS can be depleted at the intact protein level, permitting downstream top-down or bottom-up MS analysis. It must be considered, however, that intact proteins have generally poorer solubility than peptides. Thus, SDS removal from intact proteins presents a greater risk of sample loss. Intact proteins (BSA and liver extract) were also subject to KCl treatment to remove SDS, employing identical conditions to the peptide-level depletion. As shown in Figure 1B, pH had a far more pronounced impact on protein recovery. Notably, at pH 3, BSA is completely removed from the solution, due to co-precipitation with SDS + KCl. Modest recovery of liver proteins was observed in acidic conditions (32 ± 5%), though half that seen at the peptide level. This higher recovery relative to BSA may be a consequence of smaller or degraded protein fragments present in the sample. An SDS PAGE image of the liver sample, depleted of SDS over a range of pH, is provided as Appendix A. At neutral pH, BSA recovery improves significantly (70 ± 6%) and is consistent with prior studies involving KCl precipitation {Formatting Citation}. Highly basic conditions lead to a further increase in the recovery of both BSA and liver proteins.

### 3.2. Variables Impacting Protein Recovery and SDS Precipitation Efficiency

The influence of pH during KCl precipitation of SDS was further examined by quantifying recovery of a highly acidic (pepsin, pI ~2.9), and basic protein (cytochrome c, pI ~9.9), relative to the more neutral BSA (pI ~5.6). Figure 2 plots the recovery of each protein over a range of pH values, together with that of the liver whole proteome extract. The charge carried by individual proteins is calculated as a function of pH and included in the figures.

As seen in Figure 2, protein recovery is negatively influenced by an increase in total positive charge. Pepsin, having the lowest cationic sites per protein, displayed the highest recovery in acidic solution (51 ± 5% at pH 3). By contrast, cytochrome c possesses a high positive charge density at pH 3 (~1 positive charge per four residues), leading to a recovery below 4%. The co-precipitation of protein with SDS also translated a net decrease in residual SDS following KCl precipitation at low pH values (72 ± 1 ppm), compared to basic (97 ± 13 ppm) or neutral conditions (94 ± 4 ppm). At higher pH, an increase in negative charge density on the protein can improve recovery by repelling SDS and suppressing co-precipitation. This is supported by the rapid rise in recovery observed over the pH range 3 to 7, even though 85 to 90% of the basic residues remain protonated over this pH range. Together, the results support that high solution pH (neutral or above) is essential to maintain protein recovery during SDS removal.

Multiple variables can affect the recovery and purity of intact proteins following KCl precipitation of SDS. For example, considering the interactions between SDS and protein, a high initial protein concentration will result in greater residual SDS (Appendix A). Diluting the sample prior to precipitation does not alter the ratio of SDS to protein. Alternatively, the ratio of KCl to SDS can be increased to maximize precipitation of the surfactant. We also explored the inclusion of urea as a protein-solubilizing additive to maintain the solubility of intact proteins. Urea is known to dissociate SDS from proteins, reducing SDS aggregation number and its concentration below its critical micelle concentration and increasing nonpolar surface exposure in micelles [11,32]. Figure 3A summarizes the protein recovery data obtained over a range of KCl to SDS ratios (all at pH 12). For all conditions examined, the inclusion of urea enhanced the solubility of intact proteins following SDS depletion, with an average increase of 8 ± 2%. However, from Figure 3B, the presence of urea also contributed to higher residual SDS. While urea is expected to weaken the interactions between SDS and protein, urea also serves to enhance the solubility of KDS. The temperature of the solution therefore has a controlling influence on KDS solubility [17], whereby precipitation at a lower temperature results in a higher purity sample (Figure 3C). The enhanced solubility of KDS is most pronounced at elevated temperatures, particularly in the presence of urea. However, temperature also influences protein recovery. We employed a yeast membrane proteome extract, solubilized using SDS, then depleted in the presence or absence of urea. Again, while urea enhances recovery, the improvement is most pronounced at lower temperatures and statistically insignificant at 30 °C (Figure 3D). From Figure 3, one notes that room temperature precipitation with urea maintains a balance of high recovery and purity. However, urea requires a corresponding increase in the relative concentration of KCl to SDS. The data of Figure 3C,D were obtained at a 10:1 mol ratio of KCl to SDS. From Figure 3A, one notes that a 20:1 ratio of KCl with urea can deplete SDS below 100 ppm, with higher recovery than a 5:1 ratio of KCl in the absence of urea. In our experience, depleting SDS well below the ‘maximal level’ of 100 ppm typically improves LC-MS/MS analysis. In summary, room temperature precipitation at basic pH, in the presence of urea and at a high ratio of KCl to SDS, provides an optimal balance of high protein recovery and purity.

An analysis of variance was conducted to determine the relative influence of pH (4.5, 8, or 12), KCl ratio (10:1 or 50:1), and urea (0 or 7.6 M) on protein recovery and purity. Table 2 summarizes the ANOVA output from triplicate measurements at each condition. While varying the pH from 4.5 to 12 has minimal impact on residual SDS, pH has the highest relative influence on protein recovery, concluding that acidic conditions are not suitable during KCl precipitation of SDS. Similarly, the KCl:SDS ratio had minimal impact on protein recovery, but the highest relative contribution to residual SDS. Urea is seen to impact both recovery and residual SDS. Of the eight sets of conditions examined, three provided a combination of high protein recovery and purity; all three involve precipitation at high pH (>8), and the inclusion of urea necessitates a higher ratio of KCl.

### 3.3. Intact Membrane Proteome Analysis by Mass Spectrometry After SDS Removal

The depletion of SDS from intact proteins can permit MS characterization by top-down methods. To that end, the current objective is to acquire the charge envelope MS spectra free of SDS adducts and at high signal to noise (S/N), consistent with best practice analysis of intact proteins [9]. Figure 4 provides the raw and deconvoluted MS spectra of three standard proteins (ubiquitin, myoglobin, ribonuclease B) following SDS removal at pH 12 by KCl precipitation. Reference spectra were obtained from the equivalent concentration of proteins, free of SDS. As seen in the deconvoluted spectra, SDS-purified samples are fully void of SDS adducts, and maintain high signal to noise. In fact, the signal intensity for each of the three proteins was greater than that of their reference spectra; myoglobin displayed a 158% signal enhancement, ubiquitin, a 168% increase and ribonuclease, a 225% improvement. Prior work with intact proteins has demonstrated that low quantities of residual SDS (<0.01%) can enhance ESI signal intensities [3]. The deconvoluted MS spectra for ribonuclease B (Figure 4E,F) reveal multiple peaks separated by 162.1 Da, indicative of glycosylated proteoforms ranging from five to nine mannose units per protein. Appendix A summarizes additional MS spectra for intact protein standards following KCl precipitation at pH 8 vs. 12, with or without 8 M urea. For these water-soluble proteins, inclusion of urea or basic pH conditions was not essential to maintain sufficient protein recovery during SDS depletion.

Thylakoids from the chloroplasts of spinach contain an array of integral membrane protein complexes involved in solar light-harvesting. These complexes comprise multiple pigment-binding proteins, such as CP29, CP26, and CP24, so-named based on their apparent molecular weight, as seen in the SDS PAGE image of Figure 5A. Spinach thylakoid membrane proteins were isolated and precipitated with chloroform/methanol/water to remove contaminating lipids [9]. The resulting pellet was completely insoluble in an aqueous buffer without detergent. The pellet was therefore solubilized in 0.5% SDS, then subject to KCl precipitation using each of the four optimized conditions. SDS PAGE as well as LC-UV of the resulting SDS-depleted samples demonstrate the relative recovery of each precipitation condition, as shown in Figure 5B. The inclusion of urea, as well as basic pH conditions during KCl precipitation, was seen to enhance the recovery during SDS depletion from these integral membrane protein components (Figure 5B).

Accurate mass measurement of membrane proteins by mass spectrometry requires adequate protein recovery and purity in an MS-compatible buffer. Maintaining solubility of membrane proteins throughout sample processing, including the SDS depletion step is inherently more challenging than for water-soluble proteins. Membrane proteins can bind a greater quantity of SDS [33], which could manifest an increase in SDS adducts, or lower S/N in the MS spectra of the intact proteins. We herein demonstrate multiple high-quality charge envelope spectra from the spinach membrane sample. While the analysis of water-soluble protein standards did not necessitate inclusion of urea of high pH conditions during KCl precipitation (Figure 4), the most favorable charge envelope MS spectra for the spinach membrane proteins were recorded with 8 M urea, and at pH 12. As shown in Figure 6, the MS spectra obtained from KCl precipitation at pH 8 (shown at left in blue) were generally lower in intensity and/or contained SDS adducts. Raising the pH to 12 during SDS removal generally resulted in higher intensity MS spectra (green spectra at center of Figure 6), although prominent SDS adducts (+266 Da) were observed. The SDS adducts remain attached to the protein during reversed-phase LC separation ahead of MS, illustrating the greater challenge of removing tightly bound detergent compared to simple desalting protocols. However, using KCl precipitation, the combination of a highly basic solvent with 8 M urea during SDS removal provided clean MS spectra with the highest S/N ratio.

The accurate mass of the distinct proteins is an indication of unique proteoforms. For example, comparing the spectra shown in Figure 6A–C, two protein peaks were observed at similar mass, 10,230.3 Da and 10,246.3 Da, a difference of 16.0 Da, which is indicative of a single oxidation of the protein. Employing the pH 12 condition in the absence of urea (Figure 6B), prominent SDS adducts are observed, corresponding to spacing of 266 Da in the deconvoluted MS spectrum. The presence of these adducts complicates the interpretation of the charge envelope spectrum and reduces signal to noise. Incorporating urea in the KCl precipitation experiment substantially reduced the prominence of these SDS adducts (Figure 6C).

While the precipitation of SDS at pH 12 provides higher total recovery of spinach proteins, the residual presence of tightly bound SDS can have detrimental effects on the quality of the MS spectrum for intact proteins. A comparison of Figure 6J–L highlights the detection of a spinach protein with intact mass 9664.9. SDS precipitation at pH 12 with urea resulted in an approximate 10-fold improvement in signal intensity relative to SDS removal at pH 8. Moreover, the absence of urea at pH 12 (Figure 6K) completely suppressed detection of this protein. Although each of the four SDS depletion protocols may prove beneficial for favorable purification of specific proteins from a proteome system, the data suggest that KCl addition at a high pH condition with urea is most suited for membrane proteome analysis. A more detailed analysis of purified membrane proteins is next performed using bottom-up proteomics approaches.

### 3.4. Bottom-Up Membrane Proteome Analysis Following SDS Precipitation

Purification of intact proteins by SDS precipitation with KCl has previously been demonstrated, though no study has applied this approach for MS-based proteomics, particularly for a membrane proteome system. As shown here, the precipitating conditions can be tuned to balance protein recovery and purity, potentially facilitating multiple routes for successful MS analysis. Proteins from a liver membrane preparation were extracted directly into 2% SDS, ensuring only detergent-soluble proteins were recovered. While removal of the SDS does not necessarily reverse their solubility, certain proteins may be differentially impacted. MS analysis is herein employed to assess the utility of various KCl precipitation protocols. Though bottom-up MS was employed, we note that a top-down approach could also be implemented to characterize purified intact proteoforms.

A two-level, three-factor design was employed, whereby the pH (8 vs. 12), KCl ratio (10:1 vs. 50:1) and urea (0 vs. 8 M) were altered, arriving at eight unique precipitation conditions (each in duplicate). Figure 7A summarizes the protein recovery and residual SDS in the resulting samples. The initial protein concentration was high (1 g/L), permitting a five-fold dilution prior to trypsin digestion, which therefore lowers SDS by a factor of 5 as well. We selected four KCl processing conditions for bottom-up proteomics analysis (labelled I through IV in the figure), with proteome recovery from the liver membrane preparation ranging from 58 to 85%. Additionally, each sample contained a maximum 30 ppm SDS (0.03%) during enzymatic digestion. MS analysis was conducted for each duplicate sample. Figure 7B,C provide comparisons of the unique proteins and peptides identified through MS-based proteomics. Additional comparison of proteins from various sample preparation conditions (low vs. high pH; no urea vs. 8 M urea) are provided in Figure 4D,E while Figure 4F emphasizes a comparison of condition (I) to (II) (pH 8 without vs. with urea).

An analysis of the MS data reveals unique proteins identified from the four SDS precipitation workflows. Expectedly, certain conditions provided a preferential higher number of identifications, though not necessarily as predicted based on protein recovery. Merging all data sets, 1826 unique protein groups were identified, whereby 58% to 76% of the total proteins were observed in any one of the four sample workflows. The 777 proteins common to all four conditions constitute 43% of the total proteins observed. The condition with the highest number of unique protein groups, 1453, was condition (II), representing precipitation at pH 8 with urea. Condition (IV), pH 12 with urea, lead to the lowest number of unique identifications (1052). The higher protein counts obtained in condition (II) are echoed in the peptide level data (Figure 4C), whereby increased protein counts correlate with higher peptide IDs (R^2^ 0.59). From 10,029 unique peptides observed, 67% were identified in (II), while only 44% were seen in (IV).

Similar residual SDS was obtained in all conditions. Focusing on the conditions with lowest and highest proteins IDs, 26 ± 1 ppm residual SDS was present for (II) while 30 ± 1 ppm was seen in (IV), suggesting that protein purity was not a major contribution to the resulting MS data. Moreover, all samples were assayed following LC-UV cleanups (post-digestion) and contained a maximal 5 ppm SDS at the time of their introduction to the LC-MS platform. This low residual SDS would have no impact on MS analysis. Thus, protein recovery becomes the default explanation for the differences in proteomic data. However, perhaps contrary to expectations, condition (IV), with the lowest protein counts, showed the highest recovery at 87 ± 10% yield, while (II), the highest protein counts, achieved 67 ± 2% yield. It should be emphasized that a ~30% increase in recovery (and thus, a corresponding 30% increase in mass injected into the MS instrument) should not correlate with a 30% increase in unique proteins. A more important consideration is the recovery of individual proteins in the mixture, whereby the sample processing conditions can favor maintaining solubility of distinct components leading to their detection.

Pooling data from the various preparation conditions allows assessment of the impacts of pH and urea on total protein identifications. Figure 7D provides a Venn diagram examining unique proteins identified as a function of pH. While precipitation at high pH achieved higher recovery, a far greater number of unique proteins (469) were observed at pH 8, compared to 47 unique at pH 12, translating to 26% of the total proteins identified across all four conditions. From Figure 7E, precipitation in the absence of urea resulted in 326 unique proteins (19%), being more than double that of the 145 unique proteins identified when urea was included. Consider Figure 7C, the combination of urea at pH 8, with high KCl to precipitate SDS (condition II), failed to identify 328 proteins (18%). However, when considering only the low pH precipitation (pH 8), the inclusion of urea identified more unique proteins (429) than the absence of urea (328, Figure 7F).

In the context of a detergent-based workflow, a primary objective is to enhance the detection of membrane proteins. Figure 8A confirms that 44% of all characterized proteins were characterized as membrane-specific, with 17% confirmed or predicted to exhibit transmembrane components. Only 22% of all characterized proteins were classified as non-membrane components. This confirms that the SDS extraction protocol achieves a highly membrane-specific proteome fraction.

Figure 8B summarizes the total numbers of membrane and transmembrane proteins from each of the four SDS depletion workflows. Condition (IV), i.e., pH 12 + urea, which reveals the lowest number of total unique proteins, was seen to generate the greatest number of membrane (732) and transmembrane (297) proteins. This in turn contributed to a significantly higher percentage of membrane-associated proteins identified by MS, with 69.3%; furthermore, 28.1% were predicted to contain transmembrane segments. Protein GRAVY scores provide an assessment of protein hydrophobicity, with more negative scores tending towards more polar protein. Condition (I), precipitation at pH 8 without urea, generated a distribution of GRAVY scores tending towards a more negative score, which is seen by the shift in the fitted Gaussian distribution (Figure 8C) and a lower average in the Box and Whisker plot of Figure 8D. The addition of urea, increasing the pH, or the combination of both variables resulted in a more positive GRAVY score distribution for the identified proteins.

## 4. Discussion and Conclusions

Incorporating SDS in a proteomics workflow can prove beneficial under two strict conditions: First, the surfactant should only be added when necessary to extract and solubilize the target proteins. Second, the concentration of SDS must be sufficiently low at the time of downstream processing so as not to hinder the analysis. Enzymatic digestion, liquid chromatography separations, and electrospray ionization mass spectrometry are all negatively impacted by the presence of SDS. While each approach tolerates a certain level of SDS, the benefits of incorporating surfactant to improve protein solubility would be lost if the SDS concentration remains too high. A concentration of 0.01% SDS (100 ppm) is typically cited as the maximal permissible level of SDS to permit LC-MS analysis; lower surfactant concentrations are desired for optimal detection. For comprehensive proteoform analysis, MS analysis of intact proteins (i.e., via top-down approaches) is essential, therefore necessitating SDS removal at the intact protein level. Of the many approaches available for SDS depletion (Table 1), only some are suited for intact protein purification.

In this study, a membrane proteome sample was solubilized in SDS and subject to purification by KCl precipitation of the surfactant. The addition of SDS at an initial concentration well above the CMC (here, 2% SDS) was essential to solubilize proteins from the membrane pellet of spinach chloroplasts or liver, thus justifying the use of surfactants and necessitating an SDS depletion protocol. The precipitation of SDS by KCl has not previously been validated in a proteomics workflow at the intact level, nor has it been employed previously to remove SDS from membrane proteoforms. Consistent with other reports, we confirm that low pH conditions maximize the interactions between SDS and protein; the consequence of this is that intact proteins co-precipitate with the potassium detergent, leading to low recovery. Adduct formation between SDS and intact proteins is higher for hydrophobic proteins, which adds to the challenge of SDS removal. Low molecular weight, hydrophilic proteins, and digested peptides are less impacted by pH, although recovery is significantly higher when precipitation is conducted at neutral or basic conditions. Highly basic conditions are typically employed in a membrane proteome preparation, to weaken the association between proteins and lipids [28]. Precipitation of SDS with KCl has an equivalent effect in weakening SDS–protein interactions, thus minimizing the potential for co-precipitation. The recovery data provided here clearly support highly basic conditions (pH 12) to maximize protein recovery during KCl-induced precipitation of SDS.

The incorporation of urea has previously been demonstrated in an SDS-depletion workflow, most notably with FASP, and as a means of weakening the association of SDS and protein. Our hypothesis was that urea would act as a substitute for SDS following depletion, thus maintaining protein solubility. Unfortunately, urea also serves to increase the solubility of the potassium salt of dodecyl sulfate. In other words, the potential benefits of urea seen with FASP in improving protein purity are lost with KCl precipitation; urea lowers the sample purity. To that end, other parameters of the precipitation workflow must be optimized to ensure sufficient protein purity. Here, the combination of temperature and KCl to SDS ratio were optimized—higher KCl at a lower temperature provides the greatest protein purity. Thus, it is essential to consider both the protein recovery and purity as interlinked variables when optimizing an SDS depletion protocol.

Although the inclusion of urea, and highly basic conditions, provide favorable protein recovery, SDS depletion by KCl precipitation at pH 8, and/or in the absence of urea, can still achieve samples of sufficient purity to warrant subsequent LC-MS/MS proteomics processing. We therefore examined the relative efficiency of four different precipitation conditions (pH 8 vs. 12, and with vs. without 8 M urea), for proteoform analysis, wherein each resulted in approximately the same level of residual SDS, though achieving variations in protein recovery ranging from 58 to 87% following SDS removal. In the context of an MS experiment, the absolute difference in mass injected into the system would not translate significant differences in identified proteins. However, differences in the relative recovery of specific proteins could translate markedly different proteomics profiles when comparing the four sample preparation conditions.

Our objective is to optimize a workflow for the identification of hydrophobic membrane proteoforms. Purification of intact proteins by KCl precipitation of SDS at pH 12, together with the inclusion of urea increased the relative efficiency of membrane proteoform characterization by MS. The higher recovery experienced under these conditions is therefore attributed to membrane-specific proteins. Pathway analysis of those proteins uniquely identified in the urea-containing samples revealed a strong association with mitochondrial proteins (9% by cellular pathway analysis) as well as those of the endoplasmic reticulum (9%)—both membrane-rich organelles. By comparison, proteins unique to the urea-free samples were most strongly associated with the cytoplasm (25%), nucleus (19%), and cytosol (18%). The identical trends were seen comparing unique proteins at high pH (17% mitochondrion) vs. low pH (23% cytoplasm, 17% cytosol, and 8% nucleus). The presence of more polar cytosolic and nuclear proteins in the low pH, urea-free samples would suppress the detection of otherwise hard-to-digest or hard-to-ionize membrane protein components. The inclusion of SDS favors solubilization of otherwise difficult-to-extract membrane-embedded proteoforms. This class of proteins is often considered the most important class of proteins in the context of biomarker discovery, and probing drug–target interactions, despite being underrepresented proteoforms from complex biological samples. Thus, it is essential that the sample preparation workflow maximizes the solubility of the most difficult to detect proteins, to permit their optimal detection.

As shown here, KCl precipitation of SDS in the presence of urea, at high pH, is presented as a simple and favorable approach to membrane proteome processing by MS. Several high-quality charge state envelope MS spectra were recorded following purification of spinach thylakoid membrane proteins. The 16 Da mass difference of co-eluting proteins suggests detection of proteoforms (oxidation), although it is unknown if this is a biproduct of sample preparation. Obtaining these spectra with high signal-to-noise, free of SDS adducts, enables subsequent characterization by tandem MS approaches. This work further presents a bottom-up MS analysis of the purified liver membrane proteins, which is therefore limited to fully characterize proteoforms, including their modifications. Future work will be directed at a top-down MS/MS analysis of the resulting KCl-purified samples, which would prove useful for detailed proteoform characterization and relative quantitation. The utility of this sample purification workflow for characterization of clinically relevant tissues and proteomic mixtures will also be described in future studies.

## Figures and Tables

**Figure 1 proteomes-13-00030-f001:**
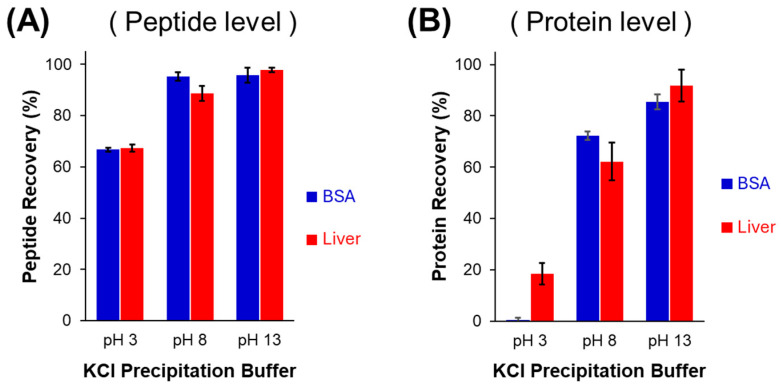
Sample recovery following SDS removal by KCl precipitation at various sample pH values, as determined by LC-UV. (**A**) SDS depletion at the peptide level, following trypsin digestion of BSA and of liver (1.0 g/L peptide with 0.5% SDS; 50:1 KCl to SDS). (**B**) SDS removal from intact proteins, under equivalent conditions. Error bars show standard deviation from three replicates.

**Figure 2 proteomes-13-00030-f002:**
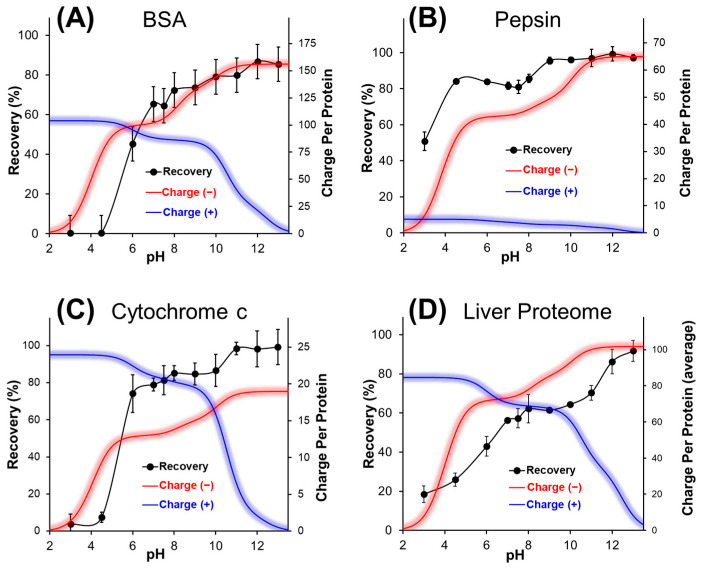
Recovery of individual proteins (**A**–**C**), and of a liver total proteome extract (**D**), following KCl addition and SDS precipitation from intact proteins. The charge of the individual proteins is calculated as a function of pH based on the numbers and associated pKa of acidic and basic residues per protein.

**Figure 3 proteomes-13-00030-f003:**
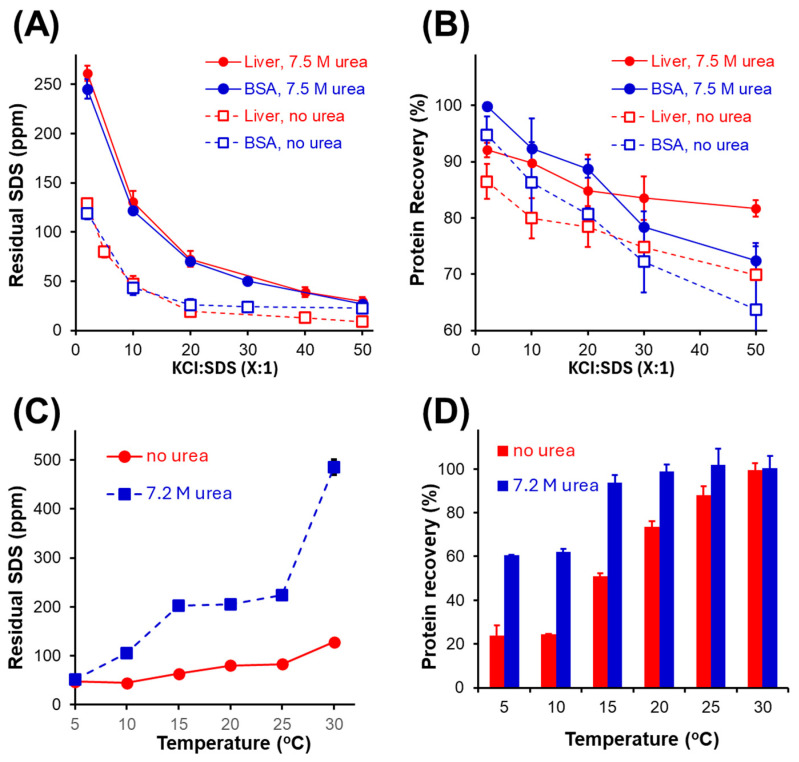
Impact of KCl:SDS ratio and urea on (**A**) protein recovery and (**B**) residual SDS following KCl precipitation of the surfactant at pH 12. Intact protein was prepared at 0.1 g/L with 0.5% SDS. A yeast membrane proteome extract was employed to examine the influence of temperature on (**C**) residual SDS and (**D**) protein recovery in the presence or absence of urea. Error bars are standard deviation from three independent samples.

**Figure 4 proteomes-13-00030-f004:**
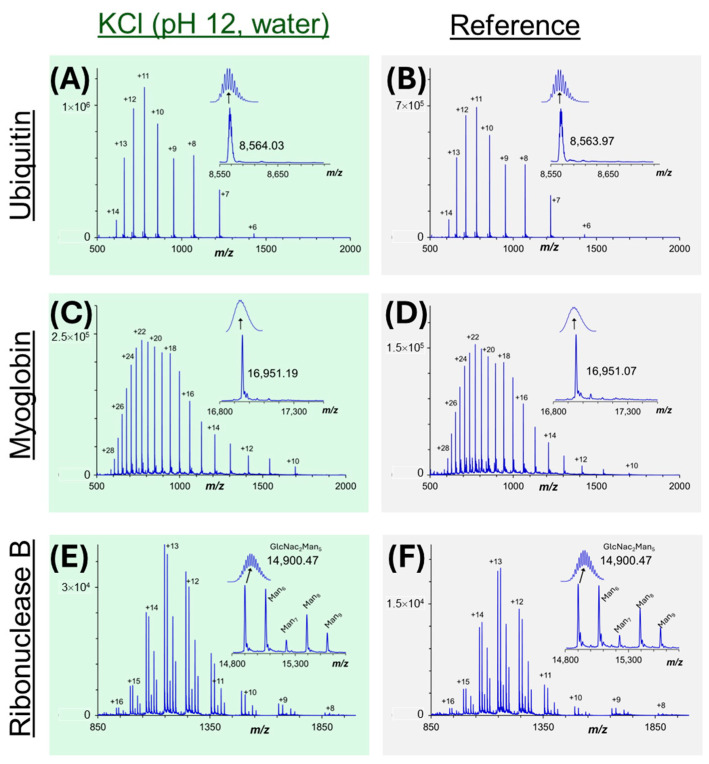
Intact MS spectra for three standard proteins indicated at the right of the image are shown together with the deconvoluted MS spectra (inset). An expanded view of the deconvoluted protein peak is also provided (resolution ~10,000 FWHM). Panels (**A**,**C**,**E**) follow the KCl depletion of SDS (initially 0.5%), while (**B**,**D**,**F**) show MS spectra collected for these proteins at equivalent concentration, in the absence of SDS. The relative intensity and absence of SDS adducts demonstrate high recovery, high purity samples.

**Figure 5 proteomes-13-00030-f005:**
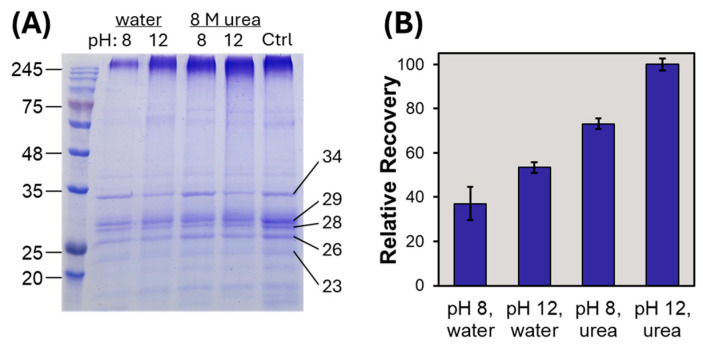
(**A**) SDS PAGE of spinach thylakoid membrane samples following SDS depletion by KCl precipitation (conditions indicated above each gel lane), relative to a control sample (Ctrl) representing the chloroform/methanol/water pellet solubilized with gel loading buffer and equivalent loading by assuming 100% recovery throughout SDS depletion. The calculated molecular weight of prominent bands are listed to the right of the gel image. (**B**) The relative recovery of proteins remaining in solution following detergent removal with the four KCl precipitation conditions was determined by LC-UV.

**Figure 6 proteomes-13-00030-f006:**
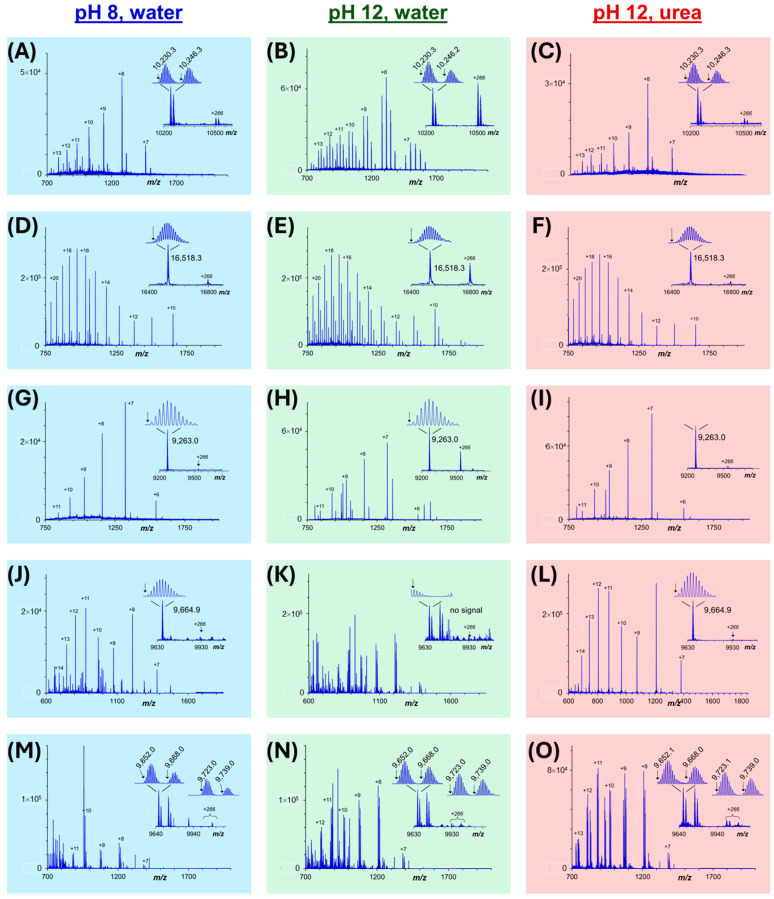
Samples of charge envelope MS spectra recorded following analysis of spinach thylakoid membrane proteins, depleted of SDS by KCl precipitation at pH 8 (blue spectra at left, (**A**,**D**,**G**,**J**,**M**)), pH 12 (green spectra, center, (**B**,**E**,**H**,**K**,**N**)), or at pH 12 with 8 M urea (red spectra, right, (**C**,**F**,**I**,**L**,**O**)). The deconvoluted MS spectra (inset) and expanded view of the deconvoluted protein peak indicates the molecular weight of the monoisotopic peak(s).

**Figure 7 proteomes-13-00030-f007:**
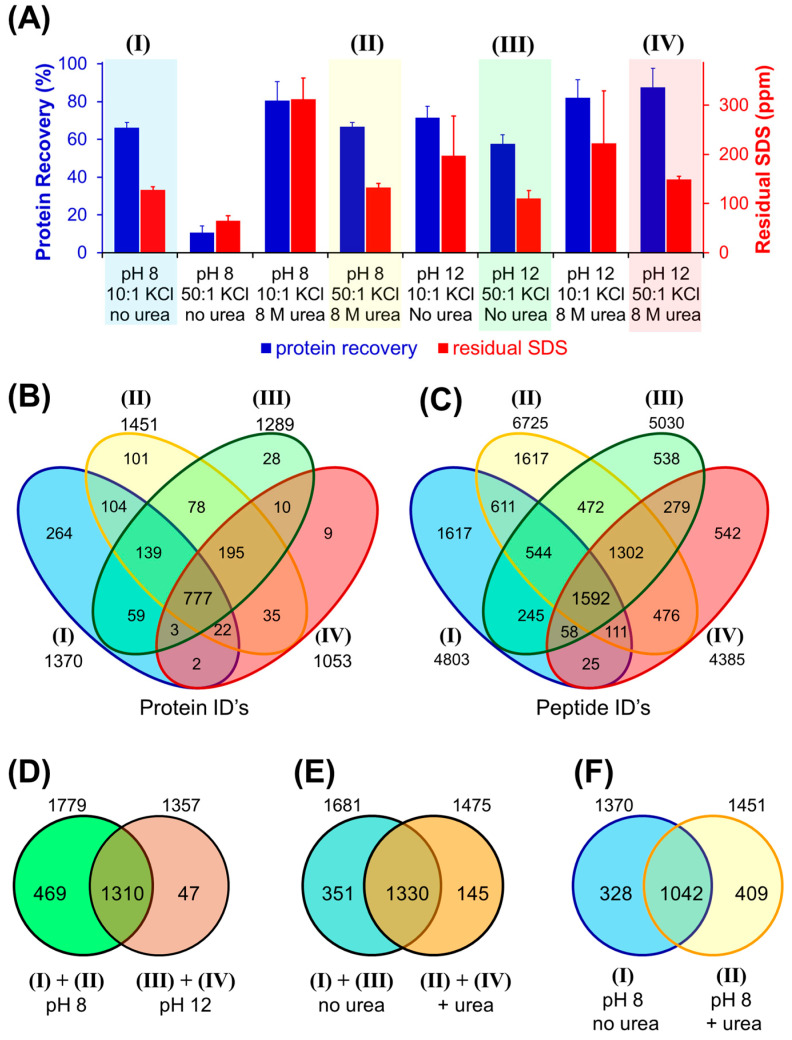
(**A**) Recovery and residual SDS following KCl precipitation of surfactant for a liver membrane proteome extract, performed at the intact protein level (2 g/L protein with 2% SDS initial). The four sample conditions subject to MS-based proteomics are color-coded and labelled I through IV. Venn diagrams summarize (**B**), the unique proteins and peptides, (**C**), identified by MS. Additional comparisons of protein IDs from combined sample processing conditions are shown in (**D**–**F**). The total numbers of proteins/peptides are indicated outside of the Venn diagrams.

**Figure 8 proteomes-13-00030-f008:**
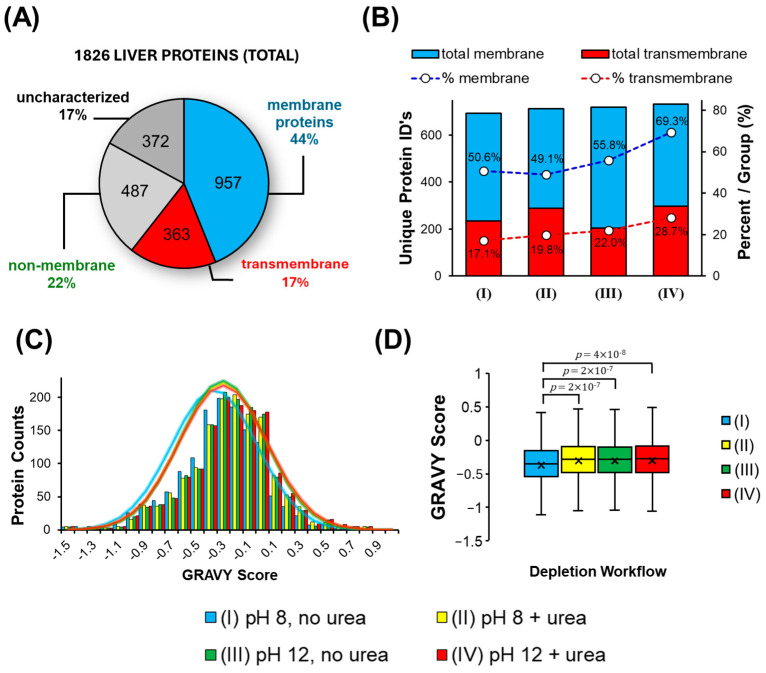
(**A**) Gene ontology analysis of cellular locations confirms most proteins are membrane-specific. (**B**) The number of membrane and transmembrane proteins identified from each of the four SDS depletion workflows, together with percentages per condition. (**C**) Histogram showing distribution of protein GRAVY scores from each depletion workflow. Overlayed are the fitted Gaussian distributions for each condition. (**D**) Box and Whisker plots compare the distribution of protein GRAVY scores, whereby SDS depletion at pH 8 in the absence of urea identified proteins tending towards more polar (negative GRAVY scores).

**Table 1 proteomes-13-00030-t001:** Summary of commonly employed SDS depletion strategies in proteomics processing.

Strategy	Description	Applies to Protein?	Applies to Peptide?	Potential Concerns	Refs
Solvent precipitation	Pellet the proteins in organic solvent (e.g., acetone, chloroform/methanol), while the detergent remains in solution; decant.	Yes	No	Proteins must be resolubilized, lowering throughput and potentially introducing sample loss.	[7]
Filter-Aided Sample Preparation (FASP)	Retain intact proteins on molecular weight filter, adding urea to weaken SDS–protein interactions; digest proteins on filter, and elute as peptides.	Yes	No	Variable protein recovery from binding to the molecular weight filters. Slow, labor-intensive processing.	[4]
Suspension Trapping (Strap)	Acid + methanol-precipitated proteins are retained on glass filter cartridge, digested, and released as peptides.	Yes	No	Commercial cartridges (higher cost).	[5]
Single-pot, solid-phase-enhanced sample preparation (sp3)	Capture intact proteins in high organic solvent on hydrophilic interaction magnetic beads, add enzyme to digest on-bead, elute the digestion products	Yes	No	Commercial cartridges (higher cost).	[6]
Metal–Organic Frameworks (MOF)	SDS retention on metal-containing nanoporous resin.	Yes	Yes	Non-selective interactions; little quantitative data on SDS removal efficiency.	[10]
Enhanced Filter-Aided Sample Preparation (eFASP)	Substitute the urea in the conventional FASP approach with sodium deoxycholate.	Yes	No	Lengthy, labor-intensive processing.	[11]
Tube-Gel Electrophoresis	Load samples into SDS PAGE gel tubes. Conventional in-gel digestion. Extensive washing releases the surfactant; trypsin releases the peptides.	Yes	No	Lengthy processing, and risk of sample loss within the gel.	[12]
Strong Cation Exchange (SCX)	Inject peptides + SDS onto cation exchange column, retaining positive peptides in acid and washing away negatively charged SDS.	No	Yes	SDS–protein interactions suggest surfactant retention on column, leading to low purity.	[13]
KCl	Potassium dodecyl sulfate has minimal solubility in water, thus precipitating the SDS while peptides remain in solution.	Yes ^(^*^)^	Yes	SDS–protein interactions can lead to co-precipitation of proteins.	[14]
Transmembrane Electrophoresis (TME)	Protein dialysis in an electric field which forces free and protein-bound SDS to migrate through the dialysis membrane, retaining intact proteins in solution.	Yes	No	Custom device which must be manufactured for use.	[15]
Pierce^TM^ Detergent Removal Spin Cartridge (Thermo Fisher Scientific, Waltham, MA, USA)	Proprietary resin in spin-column format which captures SDS, allowing peptides to pass through the resin.	Yes	Yes	Commercial cartridges (higher cost).	[16]
ProteoSpin^TM^ Detergent Cleanup Kit (Thermo Fisher Scientific)	Proprietary resin in spin-column format which captures SDS, allowing peptides to pass through the resin.	Yes	No	Commercial kit (higher cost).	-
SDS-Out^TM^ SDS precipitation kit (Thermo Fisher Scientific)	Reagent kit to induce precipitation of SDS while retaining proteins in solution	Yes	Yes	Commercial kit (higher cost).	-

* As shown in the present study.

**Table 2 proteomes-13-00030-t002:** ANOVA from a two-level, three-factor experiment design with three replicates for variables affecting KCl precipitation with protein recovery and residual SDS as the responses.

Protein Recovery						SDS Depletion				
Variable	Coefficient	Lower 95%	Upper 95%	t-Statistic	*p*-Value	Coefficient	Lower 95%	Upper 95%	t-Statistic	*p*-Value
(A) pH	−0.0053	−0.03	0.02	−0.5	0.62	21.5	19.5	23.5	23.28	1.4 × 10^−12^
(B) KCl	−0.4	−0.43	−0.38	−38.3	2.3 × 10^−16^	−2.4	−4.3	−0.4	−2.55	0.023
(C) urea	0.115	0.09	0.14	10.9	1.6 × 10^−8^	19.3	17.3	21.3	20.9	5.9 × 10^−12^
AB	0.022	0	0.04	2.1	0.053	−2.1	−4.1	−0.2	−2.31	0.036
AC	−0.028	−0.05	−0.01	−2.65	0.018	17.9	15.9	19.8	19.36	1.7 × 10^−11^
BC	0.036	0.01	0.06	3.41	3.9 × 10^−3^	−1.7	−3.6	0.3	−1.8	0.093
ABC	0.035	0.01	0.06	3.26	5.2 × 10^−3^	−1.6	−3.6	0.4	−1.7	0.11

## Data Availability

The mass spectrometry proteomics data have been deposited to the ProteomeXchange Consortium via the PRIDE partner repository with the data set identifier PXD061512 (Project Name: Bovine Liver Membrane Preparation).

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
