# Peer review of "SDS Depletion from Intact Membrane Proteins by KCl Precipitation Ahead of Mass Spectrometry Analysis"

_proteomes, 2025, doi:10.3390/proteomes13030030_

Round 1

Reviewer 1 Report

Comments and Suggestions for Authors

The manuscript submitted by Iranpou et al. described a method that precipitates SDS from membrane protein preparations by addition of potassium salts to facilitate LC-MS analysis. The authors tested  different samples at various conditions to optimize the workflow and performed real sample analysis, which proved this method is effective for either top-down or bottom-up proteomics analysis of membrane proteins. Only a few minor issues need to be addressed before the publication this manuscript. 

  1. Table 1 (line 61) should be integrated into the discussion section. There is also another Table 1 at line 360
  2. Result 3.1, is the peptide/protein recovery an average value of all peptides/proteins or peaks from the LC-UV analysis? Details of the method need to be added.
  3. L243, the mass tolerance for precursor and fragment are miss-placed. Also, setting for peptides modifications are missing.
  4. For Figure 4, are those images directly from the raw data or they are being processed? If so, please provide the software or method to generate the images.
  5. Statistical methods/tools are missing from the 2.10 Data Analysis for Table 1 at Line 360.

Author Response

COMMENT 1: Table 1 (line 61) should be integrated into the discussion section. There is also another Table 1 at line 360

RESPONSE: We thank the reviewer for catching this mistake.  Table 1 at line 360 (and discussed at line 352) has been revised to Table 2.

As per the reviewer suggestion, we added to the end of the first paragraph of the Discussion (line 573-576), emphasizing the implications of Table 1 that many SDS depletion strategies exist, but only some are suitable for intact protein manipulation.

COMMENT 2: Result 3.1, is the peptide/protein recovery an average value of all peptides/proteins or peaks from the LC-UV analysis? Details of the method need to be added.

RESPONSE:  This data refers to the total recovery of all digested peptides. These peptides all co-elute as a single peak within our LC-UV method. We clarified that LC-UV was employed in the caption to Figure 3.1.  We also clarify in the methods section (line 201) that all peptides co-elute as a single peak (it was only stated previously that all peptides are collected as a single fraction, but in fact, there is only one peak to collect, given the rapid step-wise increase from 5 to 50% which elutes all components at once).

COMMENT 3: L243, the mass tolerance for precursor and fragment are miss-placed. Also, setting for peptides modifications are missing.

RESPONSE: Line 243 (MS and MS/MS mass tolerance) has been moved to earlier in this paragraph. We also added the peptide modifications used in this paragraph (Variable: methionine oxidation; Fixed: cysteine carbamidomethylation).

COMMENT 4: For Figure 4, are those images directly from the raw data or they are being processed? If so, please provide the software or method to generate the images.

RESPONSE: The figures of the deconvoluted spectra are a direct reproduction of the original raw data, but being generated in Excel by plotting all raw MS data (ie we simply plot intensity vs m/z for all the exported data points). However, the deconvoluted spectra (insets to each image) use Excel to process the data, as stated on line 250 of the methods section.  Details of this processing step were not included, but essentially segment the various regions of the MS spectra, multiply the mass by the corresponding charge state, and sum the intensity at each corresponding mass.

COMMENT 5: Statistical methods/tools are missing from the 2.10 Data Analysis for Table 1 at Line 360.

RESPONSE: We added this to section 2.10 of the methods, indicating that the regression analysis for (table 2) were performed using built in tools from Excel.

Reviewer 2 Report

Comments and Suggestions for Authors

In this manuscript, Iranpour et al. developed a novel method to remove SDS from membrane protein samples by KCl for MS analysis. The authors provided sufficient data to show that their method performs superior to previous methods. Comparing to previous versions of the manuscript, the authors supplied more data to support their method. I believe the manuscript is suitable for publication at the current state. 

I would recommend that the figure 7 to be improved, especially panels b-f since they don’t confer much information. The authors can consider changing the sizes on the venn diagram to reflect the sizes.

Author Response

COMMENT 1: I would recommend that the figure 7 to be improved, especially panels b-f since they don’t confer much information. The authors can consider changing the sizes on the venn diagram to reflect the sizes.

RESPONSE: Venn diagrams do occasionally alter the relative size of the circles to reflect the number of components. In our case, we chose to write the total number of peptides/ proteins on the outside of each circle, so as to provide the reader with an absolute quantitative assessment of our results. Adjusting the size of each (oval) in a 4-panel Venn diagram makes it difficult to align all numbers of the many quadrants, resulting in a cluttered image. The total sizes of Figures B-F were already minimized while still being able to clearly observe each number within the respective quadrant of each figure.  It would thus be difficult to make these images much smaller.

While it can be appreciated that readers may have different opinions, we respectfully disagree on the level of information conveyed in these Venn diagrams.  Had we included a simple bar diagram plotting the total number of peptides/ proteins per method, that image would not address the relative agreement/ disagreement between methods. With Venn diagrams, in addition to showing the total peptide/protein counts recorded from each sample processing condition, it is also a quick way to visualize the agreement vs uniqueness between each method. We further compared the agreement/ uniqueness for grouped variables (eg high pH vs low pH; urea vs no urea). 

Reviewer 3 Report

Comments and Suggestions for Authors

The manuscript entitled "SDS depletion from intact membrane proteins by KCl precipitation ahead of mass spectrometry analysis" by Iranpour et al. describes the optimization of a KCl-based method to remove detergents from membrane protein samples for top-down and bottom-up proteomics. While the authors confirm that this approach is not entirely novel, the comparative results and protocol optimization presented here would be useful to the field and absolutely worth publishing. It is clear the manuscript has already gone through multiple revisions, and is quite polished and comprehensive at this stage. I see nothing wrong with the methodology or the presentation of the results. My only comment relates to Figure 4 - the legend seems incomplete ("...while (B), (D) and (F)."). Other than that, I think it's good to go!

Author Response

COMMENT 1: My only comment relates to Figure 4 - the legend seems incomplete ("...while (B), (D) and (F)."). Other than that, I think it's good to go!

RESPONSE: We thank the reviewer for catching this omission and have corrected the caption to Figure 4 of the revised manuscript (these are MS spectra of 'control' proteins, prepared at equivalent concentration but in the absence of SDS)

Reviewer 4 Report

Comments and Suggestions for Authors

Authors presents an optimized method for removing SDS from membrane protein preps using KCl precipitation for MS analysis of intact proteoforms. The authors demonstrate that high pH (especially pH 12) and the addition of urea improve membrane protein recovery and reduce SDS interference during MS analysis. In addition, other parameters of the precipitation workflow were optimized to ensure sufficient protein purity including the combination of temperature, and KCl to SDS ratio. The authors provides a robust and optimized method for SDS removal using KCl precipitation at high pH with urea, enabling high-quality mass spectrometry analysis of otherwise hard-to-detect membrane proteoforms.  

Author Response

RESPONSE: No specific changes have been requested. We thank the reviewer for their positive comments.